# The Value of Ultrasound Diagnostic Imaging of Meniscal Knee Injuries Verified by Experimental and Arthroscopic Investigations

**DOI:** 10.3390/diagnostics13203264

**Published:** 2023-10-20

**Authors:** Cezary Wasilczyk

**Affiliations:** Medical Department, Wasilczyk Medical Clinic, ul. Kosiarzy 37/80, 02-953 Warszawa, Poland; wasilczyk.chirurg@gmail.com

**Keywords:** meniscal knee injury, ultrasound, sonography, arthroscopy

## Abstract

(1) Introduction: Meniscal knee injuries may develop as the result of trauma or overloading. Ultrasound imaging is an observer-dependent modality, meaning that the assessment of structural damage depends on the investigator’s experience.. None of the published papers provides a standardized method for ultrasound examination of knee menisci. The main goal of this study is to realize and standardize ultrasound imaging diagnostics of meniscal knee injuries based on individual features of ultrasound presentation and to evaluate the applicability of this modality in clinical practice. (2) Material and methods: This study consisted of two anatomical parts, including a clinical part that started with clinical examination of 50 patients with suspected meniscal knee injuries. After this we performed ultrasound examinations in patients with positive clinical test results, using sonographic confirmation for inclusion in the next stage. Finally, knee arthroscopy by two physicians in an operating room was performed, with procedures documented through photographs and video recordings, and analytic material obtained from patients in the control group documented similarly. (3) Results: In the clinical part of the study, arthroscopic examination revealed 13 longitudinal injuries (corresponding to 36% of all injuries in the group), 14 multidirectional injuries (corresponding to 28% of all injuries), 3 radial injuries (corresponding to 6% of all injuries), and 20 oblique injuries (corresponding to 40% of all injuries). The analysis of the sensitivity and specificity of the diagnostic test in terms of recognizing actual meniscal injuries on the basis of full-thickness or partial-thickness delamination, meniscal cyst oedema, and articular space stenosis revealed that the presence of at least two of these three characteristics was associated with the sensitivity of 88% and the specificity of 86% relative to the number of actual meniscal injuries as seen in arthroscopic examination. (4) Conclusions: Research results confirm that clinical examination combined with ultrasound imaging is a very efficient tool for evaluation of meniscal injuries.

## 1. Introduction

Meniscal knee injuries may develop as the result of trauma or overloading. Meniscal pathologies are the cause of further intra-articular injuries developing in a cascade-type mechanism [1]. Traumatic injuries of the menisci can be classified as either sports-related or non-sports-related. Currently, contact sports such as football, handball, and volleyball are considered the main origins of sports-related meniscal tears. Regarding non-contact sports, those most frequently associated with meniscal injuries include skiing and acrobatics. Non-sports-related causes of meniscal injuries include traffic injuries, workplace injuries, and injuries from everyday activities [1,2,3,4,5].

The incidence of knee injuries among athletes was documented in a study carried out by Majewski and Habelt over a period of more than 10 years in a group of 17,397 patients with a total of 19,530 injuries. A total of 7769 knee joint injuries, corresponding to 68.1% of all injuries, was reported in a group of 6434 patients. Anterior cruciate ligament injuries were reported in 20.3% of cases; medial meniscal tears and lateral meniscal tears accounted for 10.8 and 3.7% of all knee injuries, respectively [3]. Medial meniscal tears are approximately 3 times more common than lateral meniscal tears [6,7]. Right knee injuries are more common than left knee injuries regardless of patients’ gender [8]. Based on the statistics of injuries suffered by Iranian football players, internal knee injuries were much more common in cases of contact-related injuries as compared to non-contact-related injuries. Most frequently, the direct mechanism of non-contact injuries involved improper ball kicks and improper landings after jumping up [9,10]. The incidence of traumatic meniscal knee injuries is estimated at 60 to 70 per 100,000 injuries in the US and at 20 per 100,000 of injuries in Norway. In the US, a total of about 850,000 meniscal tear repairs are carried out each year, with partial meniscectomy being the most common procedure in orthopedic surgery. In the UK, meniscal injuries are the cause of 25,000 hospitalizations per year. Epidemiological data on meniscal injuries differ in terms of types of injuries for subjects of different age, gender, performance status, and frequency of sporting activities. The frequency of meniscal injuries is higher in males than in females; the ratio is estimated at about 4:1 [5]. The medial meniscus is affected more frequently as compared to lateral meniscus, the ratio being estimated at 3:1. The greatest incidence of meniscal lesions in men is observed in the 2nd and the 3rd decade of life while remaining at a stable level starting from the 2nd decade of life in women. Before the age of 30, longitudinal tears, either stable or unstable, are most common in both male and female subjects. An unstable tear is defined as a tear with the damaged meniscal fragment being dislocated within the joint. A meniscal tear is considered potentially unstable if dislocation may occur within the joint under the action of external forces [11,12,13,14].

Ultrasound imaging in an observer-dependent modality means that the assessment of structural damage depends on the investigator’s experience. An experienced investigator may take note of more details and thus achieve better diagnostic results. Numerous studies assessing the efficacy of ultrasound imaging as a tool for diagnosing meniscal injuries are available in the literature [15,16,17,18,19]. None of the published papers provides a standardized method for ultrasound examination of knee menisci.

In addition to sonography, magnetic resonance imaging (MRI) is one of the most common modalities used for evaluation of meniscal morphology. MRI has an established position in the assessment of intra-articular injuries, and its diagnostic efficacy in meniscal injuries is estimated at 90–98%. A significant number of studies confirmed the high sensitivity and specificity of MRI in the diagnostics of meniscal injuries [20,21,22].

The main goal of this study is to realize and standardize ultrasound imaging diagnostics of meniscal knee injuries based on individual features of ultrasound presentation and to evaluate the applicability of this modality in clinical practice.

The objective of the experimental part of the study was to identify the sonographic features that are important for the diagnostics of meniscal damage. The objective of the clinical investigation of the study was to assess the sensitivity and specificity of ultrasound scans in the diagnostics of meniscal tears.

## 2. Material and Methods

### 2.1. Study Overview

The presented study consisted of two investigational (anatomical) parts and one clinical part. The study was approved by the Bioethical Committee in Warsaw, Poland (No. 52/21). All procedures were performed in compliance with the Declaration of Helsinki.

Part I of the anatomical study was carried out using 12 formalin-fixed specimens of normal knee menisci. The anatomical specimens were obtained from 6 lower limb specimens from deceased patients. A total of 6 lateral menisci and 6 medial menisci were obtained.

Part II of the anatomical study was carried out using 30 specimens of normal knee menisci. The anatomical specimens were obtained from 20 deceased patients. A total of 20 medial menisci and 10 lateral menisci were obtained.

The material for parts I and II was obtained from the collection of the Department of Descriptive and Clinical Anatomy of the Medical University of Warsaw.

In the clinical part of the study, the material consisted of 50 patients presenting with meniscal knee injuries and qualified for arthroscopic treatment. The inclusion criteria for patients within the study group were (1) traumatic etiology of meniscal tear; (2) positive meniscal tests in clinical examination; and (3) meniscal injury confirmed in ultrasound presentation. The study group consisted of 38 male and 12 female patients aged between 18 and 45 years. The control group in the clinical part of the study consisted of 50 patients (34 males and 16 females) aged 18 to 50 years who were subjected to arthroscopic procedures due to injuries other than meniscal tears. The clinical study was conducted at the Department of Orthopedics, Pediatric Orthopedics and Trauma Surgery of the Centre of Postgraduate Medical Education in Otwock.

### 2.2. Part I of Experimental Study

During the first stage of this part, 12 formalin-fixed specimens of normal knee menisci were obtained. The anatomical specimens were obtained from 6 lower limb specimens from deceased patients (4 men and 2 women). A total of 6 lateral menisci and 6 medial menisci were obtained. Knee specimens without meniscus damage in ultrasound examination were subjected to arthrotomy so that the normal presentation of the menisci could be verified in real life. Absence of meniscal injuries was the criterion for the use of specimen for further examination. Afterwards, longitudinal, radial, oblique, and multidirectional tears were performed in the specimens according to predefined patterns [23,24,25]. Finally, the mechanically damaged meniscus was re-examined and reassessed by ultrasound imaging.

The study was carried out by 2 sonographists who performed the evaluation of meniscal injuries and had no knowledge of the mechanism of a particular injury or the actual morphology of meniscal tear. Each trauma pattern was generated in triplicate.

Ultrasound evaluation was standardized and performed using a 12 MHz linear transducer. Images of ultrasound scans were stored on a hard drive integrated with the ultrasound system.

### 2.3. Part II of Experimental Study

A total of 30 human knee specimens were used for the study and ultrasound evaluation. The anatomical specimens were obtained from 20 deceased patients (15 men and 5 women) with mean age of 38.5 years. A total of 20 medial menisci and 10 lateral menisci were obtained.

During the first stage of this part, ultrasound examinations were performed and the knee menisci were assessed on the basis of their ultrasound presentation. Absence of injuries and unremarkable sonographic presentation were inclusion criteria to use a specimen in further study. Next, knee arthroscopy was performed using the anterolateral and anteromedial access points and the lack of meniscal pathologies was verified in actual arthroscopic images. Afterwards, meniscal injuries were generated according to predefined patterns using a surgical scalpel and arthroscopic instruments. Finally, injured menisci were subjected to reexamination and reevaluation by ultrasound imaging.

The study was carried out by 2 sonographists who performed the evaluation of menisci and had no knowledge of the mechanism of injury or the actual morphology of meniscal tear.

Types of injuries performed in meniscal specimens (morphology and location of tears) strictly reflected the patterns of traumatic meniscal knee injuries according to the current knowledge. Longitudinal, radial, oblique, and multidirectional injuries were identified. Ultrasound examinations were performed as described for the first part of the experimental study.

### 2.4. Part III—Clinical Investigation

During the first stage, a clinical examination was performed and summarized in all patients with suspected meniscal knee injuries. Next, ultrasound examinations were carried out in patients with confirmed symptoms and positive clinical test results [26,27,28]. Sonographic confirmation of meniscal knee injury in patients with positive clinical test results was used as a criterion for inclusion for the next stage of the study. At the third stage, knee arthroscopy was performed in the operating room.

Patient examinations, including clinical and ultrasound evaluation, were performed by two physicians: an experienced clinician and an experienced sonographist. The sonographist took no part in the clinical examination. Patients who qualified for surgical treatment were subjected to ultrasound examination just prior to the surgical procedure.

Ultrasound examinations were performed as described for anatomical specimens. Surgical procedures were performed within the operating room of the Department of Orthopedics, Pediatric Orthopedics and Trauma Surgery of the Centre of Postgraduate Medical Education in Otwock. Patients were placed in the supine position with sterile preparation and draping of the surgical field. Knee arthroscopy was performed under subarachnoid anesthesia from the anterolateral and the anteromedial access points [11,12,13,24,29,30]. The procedure was simultaneously documented using photographs and video recording.

Analytic material obtained from patients within the control group were documented in the form of arthroscopic photographs, video recordings, and ultrasound scans.

### 2.5. Ultrasound Imaging

All evaluations were performed using a Mylab 25 Gold system (Esaote, Italy) with 12 MHz linear transducer having 44 gain and 4.5 depth. 

Ultrasound imaging of medial and lateral menisci involved visualization of three elements: anterior horn, meniscal body, and posterior horn. With the patient in the supine position, the transducer was applied along the sagittal plane transversely to the longitudinal meniscal axis.

Projections longitudinal to the long axis of the meniscus were also used in the imaging techniques. They may be useful in the diagnostics of marginal or pericapsular zone injuries. Longitudinal cross-section of the meniscus is much less frequently used in ultrasound imaging compared to the transverse cross-section.

### 2.6. Measurements

Ultrasound presentations were assessed in terms of conformity with actual meniscal lesions. Sonographic assessment was expressed using individual scores assigned to each parameter of the injury as well as the overall score. Full-thickness delaminations detected in ultrasound images were assigned the score of 2, whereas partial-thickness delaminations were assigned the score of 1. A complete change in the meniscal morphology as observed in ultrasound scans was assigned the score of 2, whereas an incomplete change in morphology was assigned the score of 1. The presence of each of the features such as pericapsular oedema, altered morphology of coronary ligament bone attachments, and articular space stenosis was assigned the score of 1.

### 2.7. Statistical Analysis

The compliance between the ultrasound-based diagnosis and actual meniscal examination was assessed using the kappa coefficient and marginal distribution tests. The non-parametric Wilcoxon signed rank sum test was used to assess whether the total score for a particular injury was significantly different from 0, which corresponded to a non-injured meniscus.

Correlation between actual meniscal injury, as confirmed by arthroscopic examination and the ultrasound diagnosis, was evaluated in a quantitative manner.

The evaluation of individual components of the ultrasound image as risk factors for meniscal damage in patients undergoing surgery, as well as in control group patients subjected to arthroscopic examination, was carried out using a multifactorial logit regression model.

Factors taken into consideration in the model included the elements of ultrasound presentations, including delamination, change in morphological shape, capsular attachment, bone attachments, and articular space stenosis. Correlations between the statistically significant factors with the presence of meniscal injury were summarized using odds ratios (ORs) with 95% confidence intervals (95% CIs). The significance level was set at 5%. Calculations were carried out using the Stata software (v.18.0).

## 3. Results

### 3.1. Part I of Experimental Study

In part I of the experimental part of the study, ultrasonographic types of meniscal injuries were identified based on the analysis of morphological and ultrasonographic presentations of meniscal tears (Figure 1 and Figure 2). The following features of meniscal injuries as presented in ultrasound scans were identified and considered in the further part of the study: (a) the change in meniscal shape with concomitant degeneration of meniscal structure; (b) the change in the presentation of capsular attachment; (c) sharpening and degenerative lesions of the bony outlines of coronary ligament attachments; and (d) articular space stenosis.

### 3.2. Part II of Experimental Study

The following parameters of meniscal knee injuries were considered in the experimental part of the study: meniscal delamination, change in meniscal morphology, evaluation of capsular attachment, evaluation of bone attachments, and articular space stenosis. The following coding system was established for meniscal pathologies assessment in ultrasound examination:Delamination: 0: none, 1: partial thickness, 2: full thicknessMorphological change in meniscal shape: 0: none, 1: incomplete, 2: completeEvaluation of capsular attachment: 1: meniscal cyst swelling; 0: no swellingEvaluation of bone attachments: 1: degenerative lesions present; 0: unremarkableArticular space stenosis of 1 mm against a comparator: 1: present, 0: absent

The following coding system was established for meniscal pathologies assessment in arthroscopy view:0: unremarkable meniscus1: present damage

The prevalence and severity of individual pathologies in part II of the experimental study are included in Table 1.

In the pooled analysis, a total of 30 meniscal injuries were generated within the available specimens, including 5 longitudinal tears, 14 oblique tears, 5 radial tears, and 6 multidirectional tears (Table 2). As verified by an experimental study, no errors occurred in ultrasound-based identification of radial and multidirectional injuries. Longitudinal injuries were recognized correctly in four out of five specimens. In one case, longitudinal injury (20% of the test material) was misidentified as radial injury. Oblique injuries were recognized correctly in 13 out of 14 specimens. In one case, oblique injury was classified as radial.

A very high conformity was obtained for the meniscal injury type as assessed by ultrasound vs. arthroscopy, κ = 0.9; *p* < 0.0001. All but two cases (94%) were correctly evaluated based on ultrasound imaging.

A sample of arthroscopic vs ultrasound view estimation is seen in the Figure 3. (A) arthroscopy view: morphology of injury present (1). (B) ultrasound examination: change in shape morphology - 2; evaluation of capsular attachment - 1; evaluation of bone meniscus attachment - 0; articular space stenosis - 0; morphology of meniscus injury - 1.

### 3.3. Part III—Clinical Investigation

In the clinical part of the study, the material consisted of 50 patients presenting with meniscal knee injuries and qualified for arthroscopic treatment. The study group consisted of 38 male and 12 female patients aged between 18 and 45 years. The control group in the clinical part of the study consisted of 50 patients (34 males and 16 females) aged 18 to 50 years who were subjected to arthroscopic procedures due to injuries other than meniscal tears. From the collected material comprising the clinical part of the study, individual parameters of ultrasound presentation of patients with meniscal injuries and controls are included in Table 3.

In the clinical part of the study, arthroscopic examination revealed 13 longitudinal injuries (corresponding to 36% of all injuries in the group), 14 multidirectional injuries (corresponding to 28% of all injuries), 3 radial injuries (corresponding to 6% of all injuries), and 20 oblique injuries (corresponding to 40% of all injuries). Figure 4 presents arthroscopy and ultrasound views of a longitudinal, stable tear of the medial meniscus. (A) arthroscopy view: morphology of injury present (1). (B) ultrasound examination: change in shape morphology - 1; evaluation of capsular attachment - 1; evaluation of bone meniscus attachment - 0; articular space stenosis - 0; morphology of meniscus injury - 2.

### 3.4. Logit Analysis Results

Delamination (1 or 2 vs. 0), capsular attachment, and articular space stenosis were independently found to be significantly correlated with the presence of meniscal injury.

Patients in whom incomplete or complete thickness delamination was observed in ultrasound scans were much more likely to suffer from actual meniscal damage than individuals in whom no delamination was observed in ultrasound scans (OR = 23.3, 95% CI, 5.5, 99.4, *p* < 0.0001).

Subjects with meniscal cyst swelling (capsular attachment = 1) in the ultrasound scans were much more likely to suffer from actual meniscal damage than individuals in whom no swelling was observed in US scans (OR = 8.4, 95% CI, 2.1, 34.2, *p* = 0.003).

Subjects with articular space stenosis (pericapsular attachment = 1) in the ultrasound scans were much more likely to suffer from actual meniscal damage than individuals in whom no stenosis was observed in US scans (OR = 11.9, 95% CI = 2.7, 53.4, *p* = 0.001).

The model fit was verified using the Hosmer–Lemeshow goodness of fit test, with *p* = 0.06 (a non-significant result should be interpreted as a good fit).

For computational purposes, evaluation of bone attachments was not considered in the model. The absence of positive controls prevented the estimation of the odds ratios of meniscal injuries depending on the presence of degenerative lesions in bone attachments.

For computational reasons, the multifactorial model also failed to analyze the relationship between the change in meniscal morphology as seen in the ultrasound image and the presence of meniscal injury in arthroscopic examination.

Unifactorial analysis revealed that a complete change in meniscal morphology as observed in ultrasound scan was associated with the likelihood of meniscal injury, as detected in arthroscopic investigation, being increased by a factor of more than 300 (OR = 359.3, 95% CI, 41.6, 3102, *p* < 0.0001).

The analysis of the sensitivity and specificity of the diagnostic test in terms of recognizing actual meniscal injuries on the basis of full-thickness or partial-thickness delamination, meniscal cyst oedema, and articular space stenosis revealed that the presence of at least two of these three characteristics was associated with the sensitivity of 88% (44/50) and the specificity of 86% (43/50) relative to the number of actual meniscal injuries as seen in arthroscopic examination. The percentage of correctly classified cases amounted to 87% (87/100).

When the complete change in meniscal morphology was also included in the test, the presence of at least two of the four aforementioned features was associated with the sensitivity of 100% (50/50) and the specificity of 84% (42/50) relative to the number of actual meniscal injuries as seen in arthroscopic examination. The percentage of correctly classified cases amounted to 92% (92/100).

## 4. Discussion

An ultrasound evaluation of knee menisci is difficult due to the deep intra-articular location of the menisci and the proximity of bone tissues. A significant percentage of misdiagnosed meniscal pathologies are caused by its unreliability. MRI has an established efficacy in the assessment of intra-articular structures, including the menisci [31,32,33,34,35].

The rapidly developing techniques for MRI of articular structures may somewhat obscure the value of a medical interview and clinical evaluation [36]. In my view, the integration of ultrasound imaging with the assessment of risk factors, dynamic evaluation, and clinical examinations has the potential to enhance a clinician’s attention to specific issues. I hold the belief that magnetic resonance imaging should complement ultrasound imaging as a modality aimed at offering a more comprehensive understanding of a particular clinical problem. Regrettably, these two modalities are often misconceived as alternatives.

In the available research studies, the efficacy of ultrasound imaging in the diagnostics of meniscal injuries is estimated at 30 to 96% [19,37,38,39].

In a study by Mattli et al., carried out in 1993, the sensitivity and the specificity of ultrasound imaging in a group of 177 patients with meniscal injuries as verified by arthroscopic examinations amounted to 35% and 85%, respectively. These results did not confirm the high reliability of ultrasound imaging in the diagnostics of meniscal injuries [40].

In a study by Timotijevic et al., carried out in a group of 198 patients, ultrasound evaluation of fresh medial meniscal lesions had a sensitivity of 91% and a specificity of 80%, whereas evaluation of inveterate medial meniscal injuries had a sensitivity of 97% and a specificity of 90%. The percentage of correctly classified cases amounted to 86% [19].

The high sensitivity and specificity of ultrasound imaging in the assessment of meniscal injuries as verified by arthroscopic examination were also demonstrated in a group of 35 patients by Shetty et al. The sensitivity and specificity of ultrasound imaging were estimated at 86% and 69%, respectively. The percentage of correctly classified cases amounted to 82.6% [18].

In their study carried out in 1186 patients subjected to ultrasound imaging evaluation of knee menisci with arthroscopic verification, Richter et al. estimated the sensitivity and the specificity of the ultrasound method at 83% and 90%, respectively, for the medial meniscus, and at 58% and 98%, respectively, for the lateral meniscus. A study assessing the sensitivity of ultrasound in the diagnostics of knee meniscal injuries (as verified by arthroscopy) in 113 knee joints obtained post mortem from adult subjects returned the sensitivity values of 81% for medial meniscal injuries and 47% for lateral meniscal injuries [41].

The results of the above-mentioned evaluations of meniscal injuries did not include the identification and systematization of individual sonographic features, nor did they attempt to determine the extent of lesions. Diagnostic efficacy depended on the investigators’ experience. In summary, the available studies assessing the usefulness of ultrasound imaging, confirmed by arthroscopy, rely solely on the clinical experience of the investigators. None of the papers provided standardized methods for ultrasound examination of knee menisci, including both examination procedures and image evaluation guidelines. In addition, individual parameters of ultrasound presentation were not identified in the studies. The assessment of the meniscus was based on the investigator’s experience alone. Therefore, in this study I decided to make an attempt at standardization of ultrasound examination of knee menisci in a manner similar to that successfully employed in the Graf method of hip assessment in infants [42].

The proposed ultrasound examination procedure depending on the meniscal zone and including evaluation of individual image parameters has been provided as the result of the research.

The following independent image characteristics can be distinguished in aggregate statistical analysis:-Partial- (1) or full-thickness delamination (2);-Change in capsular attachment;-Articular space stenosis.

A significant correlation was observed between the combination of the above-mentioned parameters and meniscal injury as verified by arthroscopic examination via the sensitivity analysis (determination of the number of subjects in the study group being correctly diagnosed on the basis of the imaging scan) and the specificity analysis (determination of the number of subjects in the control group being correctly diagnosed on the basis of the imaging scan). The proposed multifactorial model of sonographic evaluation revealed that the presence of two of the three characteristics provided a sensitivity of 88% (44/50) and a specificity of 86% (43/50). The percentage of correctly classified cases amounted to 87% (87/100).

The inclusion of the complete lesion morphology with one of the above features provided a sensitivity of 100% and a specificity of 84%. The percentage of correctly classified cases amounted to 92% (92/100).

In unifactorial analysis, the individual features of the ultrasound image are associated with the following likelihoods of injury being confirmed in arthroscopic examination:(1)Full-thickness delamination—identification of full-thickness meniscal delamination (coded as delamination type 2) corresponded to 100% of correct diagnoses.(2)Partial-thickness delamination—Identification of partial-thickness meniscal delamination increased the likelihood of injury being also detected in arthroscopic examination by a factor of 23.(3)Full-thickness morphological change in meniscal shape (coded as type 2)—identification of full-thickness morphological change corresponded to 97% of correct diagnoses and increased the likelihood of injury being also detected in arthroscopic examination by a factor of >300.(4)Articular space stenosis—identification of stenosis increased the likelihood of injury being also detected in arthroscopic examination by a factor of 11.

The use of individual features of sonographic presentation and combinations of these features in multi- or unifactorial analyses facilitates high sensitivity and specificity, as well as repeatability of diagnosing meniscal injuries using the ultrasound imaging technique.

In practice, independent occurrence of two out of the three characteristics, namely the full- or partial-thickness delamination coded as type 1 or 2, pericapsular oedema, or articular space stenosis, is associated with a sensitivity of 88% and a specificity of 86%. The presence of a complete change in meniscal morphology coded as type 2 along with either one of the features from the group consisting of partial-thickness delamination coded as type 1, full-thickness delamination coded as type 2, pericapsular oedema, or arterial space stenosis is associated with a sensitivity of 100% and a specificity of 84%.

By utilizing individual presentation parameters, along with pooled and unifactorial analyses, ultrasound evaluation of the menisci confirms its high diagnostic efficacy, which is comparable to that of MRI [19,35,39].

Innovatively using meniscal presentation parameters with multifactorial or unifactorial analyses expands the potential for broader use of ultrasound imaging, which exhibits high diagnostic efficacy.

A review of available publications on ultrasound evaluation of knee menisci found no articles reporting on the assessment of meniscal injuries based on ultrasound presentations. While numerous studies confirm the effectiveness of sonographic assessment when compared to arthroscopic examination, procedural standards are lacking. None of the published articles reported the criteria used by the investigators when assessing the sonographic images of injured menisci as verified otherwise.

The study’s limitation is that performing ultrasound evaluations of the meniscus requires at least an intermediate level of experience and skill in musculoskeletal ultrasonography.

## 5. Conclusions

In conclusion, standardization of ultrasound imaging of knee menisci and reference to individual elements of sonographic presentation contribute to the increased sensitivity and specificity of ultrasound imaging. Research results confirm that clinical examination combined with ultrasound imaging is a very efficient tool for evaluation of meniscal injuries.

## Figures and Tables

**Figure 1 diagnostics-13-03264-f001:**
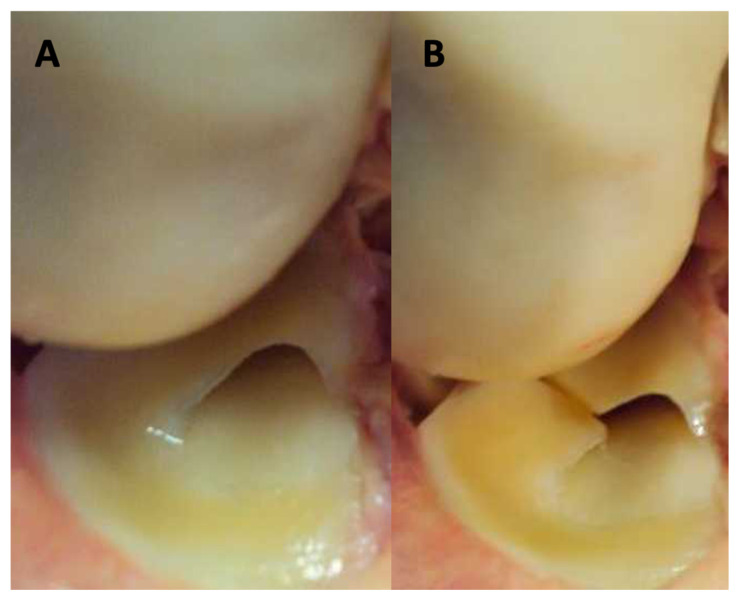
First stage of experimental study, knee joint specimen arthrotomy. (**A**) Presentation of unremarkable lateral meniscus; (**B**) presentation of damaged lateral meniscus—radial type.

**Figure 2 diagnostics-13-03264-f002:**
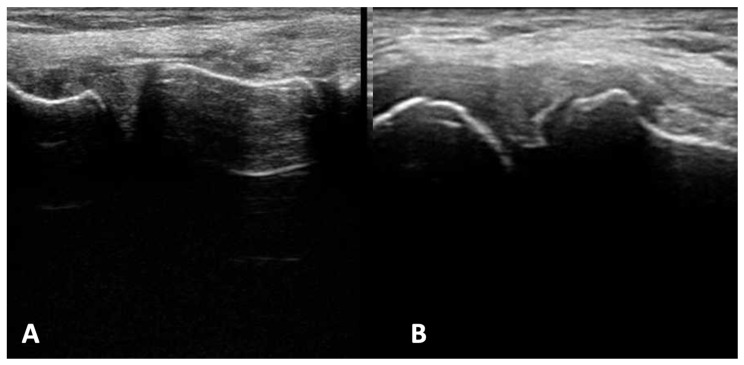
First stage of experimental study: ultrasound examination. (**A**) Presentation of unremarkable lateral meniscus; (**B**) presentation of damage of lateral meniscus—radial type.

**Figure 3 diagnostics-13-03264-f003:**
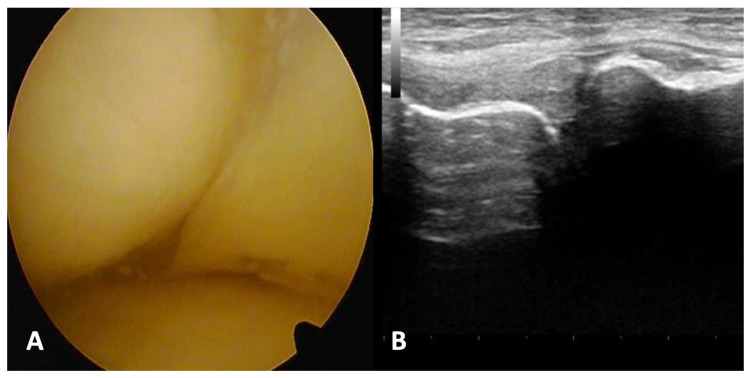
Arthroscopy and ultrasound examination results analysis of longitudinal, stable medial meniscus tear. (**A**) Arthroscopy view; (**B**) presentation of ultrasound examination.

**Figure 4 diagnostics-13-03264-f004:**
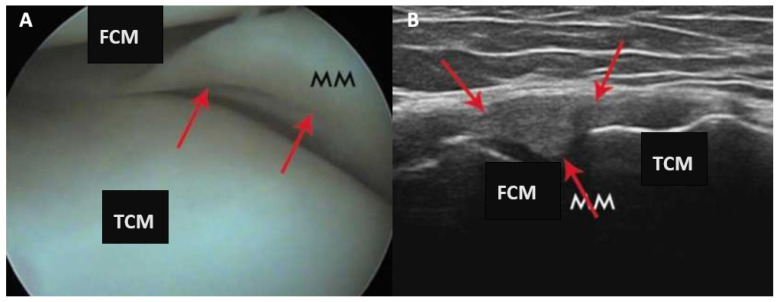
Clinical part of the study of a longitudinal, stable tear of the medial meniscus. (**A**) Presentation of arthroscopy view; (**B**) presentation of ultrasound examination. MM—medial meniscus; FCM—femur medial condyle; TCM—tibial medial condyle. Red arrows indicate damage to the meniscus in arthroscopic and ultrasound view.

**Table 1 diagnostics-13-03264-t001:** Prevalence and severity of individual pathologies in part II of the study.

Pathology		*N* = 30
Delamination	Partial-thickness Full-thickness	2 (7%) 28 (93%)
Morphological change in meniscal shape	Incomplete Complete	2 (7%) 28 (93%)
Evaluation of capsular attachment	No oedema Oedema	12 (40%) 18 (60%)
Evaluation of bone attachments	Normal Abnormal	24 (80%) 6 (20%)
Articular space stenosis	Absent Present	19 (63%) 11 (37%)

**Table 2 diagnostics-13-03264-t002:** Effectiveness of ultrasound evaluation of individual meniscus injuries based on arthroscopic verification—experimental part.

	Arthroscopic Evaluation				
Ultrasound Evaluation	Longitudinal	Oblique	Radial	Multidirectional	Total
Longitudinal	4 (13%)	0	0	0	4 (13%)
Oblique	0	13 (43%)	0	0	13 (43%)
Radial	1 (3%)	1 (3%)	5 (17%)	0	7 (23%)
Multidirectional	0	0	0	6 (20%)	6 (20%)
Total	5 (17%)	14 (47%)	5 (17%)	6 (20%)	30 (100%)

**Table 3 diagnostics-13-03264-t003:** Ultrasound parameters of patients and controls from part III of the study.

Parameters of the Ultrasound Presentation of Menisci	Arthroscopic Meniscal Injury Group, N = 50	Control Group, N = 50
Delamination 0	6 (12%)	42 (84%)
1	26 (52%)	8 (16%)
2	18 (36%)	0
Morphological change in meniscal shape 0	0	37 (74%)
1	6 (12%)	12 (24%)
2	44 (88%)	1 (2%)
Evaluation of capsular attachment 0	10 (20%)	40 (80%)
1	40 (80%)	10 (20%)
Evaluation of bone attachments 0	36 (72%)	50 (100%)
1	14 (28%)	0
Articular space stenosis 0	16 (32%)	40 (80%)
1	34 (68%)	10 (20%)

## Data Availability

The data underlying this article will be shared on reasonable request to the corresponding author.

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
