# Peer review of "The Value of Ultrasound Diagnostic Imaging of Meniscal Knee Injuries Verified by Experimental and Arthroscopic Investigations"

_diagnostics, 2023, doi:10.3390/diagnostics13203264_

Round 1
Reviewer 1 Report
I read carefully the manuscript ‘The value of ultrasound diagnostic imaging of meniscal knee injuries verified by experimental and arthroscopic investigation’ and I think that is a study with a really good experimental design and results that contribute to the use of ultrasound examination in clinical practice, however, I think that the manuscript is roughly written and not understood by the reader. The abstract is not properly structured, the introduction is missing, and many references do not exist within the text. Additionally, the pdf form of the manuscript doesn’t help in the suggestion of changes – a word form with lines is preferred. Furthermore, an explanation initially in the materials and methods about why the 3 parts of the experimental study were done and how they are connected to each other is required.
I propose an extended description of the ultrasound examination procedure (transducer placement area, longitudinal/transverse scans etc. )
Comments:
‘Ultrasound imaging and observer-dependent modality, which means that the assessment of structural damage depends on the investigator's experience’ Please rewrite to make sense
‘A significant percentage of misdiagnosed meniscal pathologies was responsible for ultrasound imaging being discarded by the orthopedic community due to its unreliability.’ Please rewrite to make sense
‘The study was carried out by 2 individuals’ the sonographer and the other?
‘Ultrasound evaluation was standardized and performed using a 12 MHz linear transducer. Images of ultrasound scans were stored on the hard drive integrated with the ultrasound system.’ Please provide more information about the ultrasound machine, the Gain, the depth, and the footprint of the transducer.
2.2. Part III - clinical investigation : ‘…. ultrasound examinations were carried out in patients with confirmed symptoms and positive clinical test results….’ Please add references about the positive clinical test.
‘Knee arthroscopy was performed under subarachnoid anesthesia from the anterolateral and the anteromedial access points…..’ Please add references about the procedure of knee arthroscopy.
‘Magnetic resonance imaging has an established efficacy in the assessment of intra-articular structures, including the menisci.’ Please add a reference.
‘In my opinion, ultrasound imaging…… . I believe that… , In this study I decided to make….. .’ please rewrite.
‘…..an attempt at standardization of ultrasound examination of knee menisci in a manner similar to that successfully employed in the Graf method of hip assessment in infant’ Please add a reference.
‘Ultrasound evaluation of the menisci taking advantage of the individual presentation parameters in combination with pooled and unifactorial analyses confirms that the diagnostic efficacy of ultrasound imaging is high and comparable with the efficacy of magnetic resonance imaging.’ Please add a reference.
Editing of English language required.
Author Response
I read carefully the manuscript ‘The value of ultrasound diagnostic imaging of meniscal knee injuries verified by experimental and arthroscopic investigation’ and I think that is a study with a really good experimental design and results that contribute to the use of ultrasound examination in clinical practice, however, I think that the manuscript is roughly written and not understood by the reader. The abstract is not properly structured, the introduction is missing, and many references do not exist within the text. Additionally, the pdf form of the manuscript doesn’t help in the suggestion of changes – a word form with lines is preferred. Furthermore, an explanation initially in the materials and methods about why the 3 parts of the experimental study were done and how they are connected to each other is required.
Authors: Thank you for this review, we have improved our manuscript according to all suggestion. All changes are marked in track changes mode. We also edited language.
I propose an extended description of the ultrasound examination procedure (transducer placement area, longitudinal/transverse scans etc. )
Authors: thank you for this suggestion, we have added section about ultrasound examination to the methods.
Comments:
‘Ultrasound imaging and observer-dependent modality, which means that the assessment of structural damage depends on the investigator's experience’ Please rewrite to make sense
Authors: thank you, we have re-writed this sentence.
‘A significant percentage of misdiagnosed meniscal pathologies was responsible for ultrasound imaging being discarded by the orthopedic community due to its unreliability.’ Please rewrite to make sense
Authors: thank you, we have re-writed this sentence.
‘The study was carried out by 2 individuals’ the sonographer and the other?
Authors: thank you, we have added an explanation.
‘Ultrasound evaluation was standardized and performed using a 12 MHz linear transducer. Images of ultrasound scans were stored on the hard drive integrated with the ultrasound system.’ Please provide more information about the ultrasound machine, the Gain, the depth, and the footprint of the transducer.
Authors: Thank you, we have added information what system we used.
2.2. Part III - clinical investigation : ‘…. ultrasound examinations were carried out in patients with confirmed symptoms and positive clinical test results….’ Please add references about the positive clinical test.
Authors: thank you, we have added references.
‘Knee arthroscopy was performed under subarachnoid anesthesia from the anterolateral and the anteromedial access points…..’ Please add references about the procedure of knee arthroscopy.
Authors: thank you, we have added references.
‘Magnetic resonance imaging has an established efficacy in the assessment of intra-articular structures, including the menisci.’ Please add a reference.
Authors: thank you, we have added references.
‘In my opinion, ultrasound imaging…… . I believe that… , In this study I decided to make….. .’ please rewrite.
Authors: thank you, we have changed entire paragraph.
‘…..an attempt at standardization of ultrasound examination of knee menisci in a manner similar to that successfully employed in the Graf method of hip assessment in infant’ Please add a reference.
Authors: thank you, we have added reference.
Ultrasound evaluation of the menisci taking advantage of the individual presentation parameters in combination with pooled and unifactorial analyses confirms that the diagnostic efficacy of ultrasound imaging is high and comparable with the efficacy of magnetic resonance imaging.’ Please add a reference.
Authors: thank you, we have added references.
Reviewer 2 Report
Dear Authors
The study title" The value of ultrasound diagnostic imaging of meniscal knee injuries verified by experimental and arthroscopic investigations" is innovative and has convincible clinical significance
Manuscript need minor English correction
Dear Editor
The study title" The value of ultrasound diagnostic imaging of meniscal knee injuries verified by experimental and arthroscopic investigations" is innovative and has convincible clinical significance
Manuscript need minor English correction
Author Response
Thank you for this review. We have re-checked our manuscript and corrected language.
Reviewer 3 Report
Thank you very much for allowing me to review this manuscript. First of all, I would like to congratulate the authors for the very interesting project they have proposed. I think there must have been some kind of error since only one author appears in the manuscript. I assume that it is an error because reading the manuscript it can be seen that it is a work done by several people. Please find my comments below.
Abstract
- Please, add the study´s aim.
- Please explain the number of patients, the type of study and the measurements taken
- The results should be better explained: How were the menisci "correctly" classified? What type of analysis was performed?
- The conclusion must be made based on the results of the study itself. Please base the conclusion on your own results based on your study objective.
Introduction
- There is no introduction!
Materials and Methods
- It would be necessary to add the ages and deviations of the cadaveric specimens.
- I cannot find the sample size calculation.
- What type of ultrasound scanner was used? Add the make and model of the equipment.
- I consider that the first part of "statistical analysis" would correspond to a section on "measurements" or "measurement procedure". I leave it to the choice of the authors.
Results
- Figure 1 (The font size and typeface is uneven).
- Very nice figures.
Discussion
- Please review the entire discussion (letters in different fonts, disconnection of the text).
Author Response
Thank you very much for allowing me to review this manuscript. First of all, I would like to congratulate the authors for the very interesting project they have proposed. I think there must have been some kind of error since only one author appears in the manuscript. I assume that it is an error because reading the manuscript it can be seen that it is a work done by several people. Please find my comments below.
Authors: Thank you for this review, we have improved our manuscript according to all suggestion. All changes are marked in track changes mode.
Abstract
- Please, add the study´s aim.
- Please explain the number of patients, the type of study and the measurements taken
- The results should be better explained: How were the menisci "correctly" classified? What type of analysis was performed?
- The conclusion must be made based on the results of the study itself. Please base the conclusion on your own results based on your study objective.
Authors: Thank you, we have corrected entire abstract
Introduction
- There is no introduction!
Authors: I do not know what version you received, but the manuscript has an introduction (Lines 61-113)
Materials and Methods
- It would be necessary to add the ages and deviations of the cadaveric specimens.
Authors: Thank you, we have provided more information as suggested.
- I cannot find the sample size calculation.
Authors: We do not perform sample calculation.
- What type of ultrasound scanner was used? Add the make and model of the equipment.
Authors: Thank you, we have added information what system we used.
- I consider that the first part of "statistical analysis" would correspond to a section on "measurements" or "measurement procedure". I leave it to the choice of the authors.
Authors: Thank you for this suggestion, we have added section measurements.
Results
- Figure 1 (The font size and typeface is uneven).
- Very nice figures.
Authors: Thank you, please recheck the font size, as it seems to be displaying correctly.
Discussion
- Please review the entire discussion (letters in different fonts, disconnection of the text).
Authors: Thank you. Please review the discussion section again, as it seems to be displaying correctly. Nevertheless, we have made some corrections.
Round 2
Reviewer 1 Report
The manuscript has been improved. I consider it suitable for publication after minor corrections.
Comments
Abstract. 'Finally, knee arthroscopy performed by two physicians in an operating room was performed,...' please correct the draft of the sentence.
Table 4. The results of logit analysis. The table is inside the text.
Moderate editing of English language required
Author Response
Abstract. 'Finally, knee arthroscopy performed by two physicians in an operating room was performed,...' please correct the draft of the sentence.
Authors: Thank you we have modified this sentence.
Table 4. The results of logit analysis. The table is inside the text.
Authors: True, we decided to delete Table 4.
Reviewer 3 Report
Nice work! The authors have improved the article.
Author Response
Thank you